# Antimicrobial-Resistant Bacteria from Free-Living Green Turtles (*Chelonia mydas*)

**DOI:** 10.3390/antibiotics12081268

**Published:** 2023-08-01

**Authors:** Fernanda S. Short, Gisele Lôbo-Hajdu, Suzana M. Guimarães, Marinella S. Laport, Rosane Silva

**Affiliations:** 1Instituto de Biofísica Carlos Chagas Filho, Universidade Federal do Rio de Janeiro, Rio de Janeiro 21941-902, Brazil; shortfernanda@biof.ufrj.br; 2Instituto de Biologia Roberto Alcantara Gomes, Universidade do Estado do Rio de Janeiro, Rio de Janeiro 20551-030, Brazil; lobohajdu.uerj@gmail.com; 3Projeto Aruanã, Instituto de Pesquisas Ambientais Littoralis, Rio de Janeiro 24320-330, Brazil; sumgbio@gmail.com; 4Instituto de Microbiologia Paulo de Góes, Universidade Federal do Rio de Janeiro, Rio de Janeiro 21941-902, Brazil; marinella@micro.ufrj.br

**Keywords:** One Health, AMR, microbial distribution, marine pollution biomonitor

## Abstract

Bioindicator species are used to assess the damage and magnitude of possible impacts of anthropic origin on the environment, such as the reckless consumption of antimicrobials. *Chelonia mydas* has several characteristics that make it a suitable bioindicator of marine pollution and of the presence of pathogens that cause diseases in humans. This study aimed to investigate the green sea turtle as a reservoir of resistant bacteria, mainly because *C. mydas* is the most frequent sea turtle species in Brazilian coastal regions and, consequently, under the intense impact of anthropic factors. Free-living green sea turtles ranging from 42.8 to 92 cm (average = 60.7 cm) were captured from Itaipú Beach, Brazil. Cloaca samples (characterizing the gastrointestinal tract) and neck samples (representing the transient microbiota) were collected. Bacterial species were identified, and their was resistance associated with the antimicrobials cephalothin, ciprofloxacin, gentamicin, tetracycline, and vancomycin. *Citrobacter braaki*, *Klebsiella oxytoca*, *K. variicola* and *Proteus mirabilis* were found resistant to cephalothin and *Morganella morganii* and *Enterococcus faecalis* tetracycline-resistant isolates in cloaca samples. In neck samples, species resistant to tetracycline were *Salmonella* sp., *Serratia marcescens*, *S. ureylitica* and *Proteus mirabilis*. This data reinforces that the green turtle is a bioindicator of antimicrobial resistance (AMR).

## 1. Introduction

The increase in antimicrobial resistance (AMR) may be due to rampant use in humans and animals. Antimicrobials are used in animals as a prophylactic measure and growth promoter since the administration of low-dose antimicrobials (sub-therapeutic use) provides rapid weight gain in animals [1]. It has been predicted that resistant bacteria will cause the deaths of around 10 million people [2], although these figures are controversial and may vary [3] depending on future scenarios [4]. Antimicrobial resistance (AMR) has been a public health issue for some time, and it has been designated a global public health priority [5]. AMR can result in fatalities, increased healthcare costs, difficulty controlling infectious diseases, a threat to health and safety, and a weakened trade and economy due to the high costs of treating these infections [6]. It is essential to comprehend and identify the connections between human, animal, and environmental microbiota [7], given the frequent dynamics of bacterial antimicrobial resistance genes (ARGs) across species and environments.

Antimicrobials belong to different classes that have their own representative drugs. They also have diverse targets, such as the cell wall and membrane, protein biosynthesis, DNA structure, and the synthesis of purines and folic acid. Mechanisms of action are inhibition of peptidoglycan synthesis present in the bacterial cell wall, as in ampicillin, cephalothin, and vancomycin; inhibition of protein synthesis, as in gentamicin and tetracycline; and inhibition of the activity of the topoisomerase II enzyme, as in ciprofloxacin [8]. Given the frequent dynamics of bacteria and ARGs across species and environments, it is crucial to understand and identify the links between human, animal, and environmental microbiota [7].

One of the major problems in controlling these microorganisms is the inappropriate disposal of antimicrobials, which end up going into the environment, contaminating food, and thus generating a cycle throughout the food chain, returning to their anthropic origin. In this context, the One Health approach can be used to balance and sustainably optimize the health of people, animals, and ecosystems. It recognizes that the health of humans, domestic and wild animals, plants, and the wider environment (including ecosystems) are closely linked and interdependent [9]. Therefore, programs, policies, legislation, and research can be designed and implemented to combat the spread of antimicrobial resistance with the help of public health, environmental, and food surveillance [10,11].

Bioindicator species are used to examine environmental health and thus be able to assess the damage that the reckless consumption of antimicrobials is causing [12]. The green turtle, *Chelonia mydas* (Linnaeus, 1758), has several characteristics, such as longevity and broad migratory patterns, that make it a suitable bioindicator of marine pollution and human health [13]. In adulthood, they alternate periods of migration offshore to reach feeding areas, returning to coastal areas in the breeding season [14]. Green turtles are globally distributed, with a concentration in the tropics, and are often considered sentinels of the ocean [13]. They spend most of their lives in coastal environments, where they are in contact with antimicrobial-resistant bacteria from humans [15]. The persistence of sea turtles in coastal areas contributes to the dissemination of these microorganisms in the ocean [16]. Furthermore, early diagnosis may help avoid unnecessary antimicrobial therapy to treat infectious diseases in sea turtles [17], avoiding the dissemination of antimicrobial resistance.

The main current threats to sea turtle populations have an anthropic origin, such as the loss of nesting areas, pollution of the seas, by catch, and other harmful practices [18]. Sea turtles play an intense role in the marine ecosystem, as they help balance food webs and are predators, controlling populations of algae, jellyfish, and sponges, among others [19]. Any alteration in the physiology or behavior of the animal, such as fibropapillomatosis (a disease caused by a virus from the herpes family) or parasites, is a warning about the health of the ecosystem [20]. 

Methods for dealing with antimicrobial resistance are laborious and intensive. Common methods include the collection of samples from sea turtles, isolatation and identification of bacteria, and subsequent use of susceptibility tests [21,22,23,24]. Various techniques are used for detecting microorganisms, such as microscopy, biochemical, and molecular methods. The Matrix-Assisted Laser Desorption/Ionization (MALDI)-Time-of-Flight (TOF) is an advanced method for identifying microorganisms that is fast and cost-effective. It is particularly useful for classifying bacteria according to their taxonomy [25]. With the arrival of molecular techniques, the identification, classification, and characterization of microorganisms have become more reliable and extensive as16S rRNA gene sequencing. Both methods are effective, and the decision to use one or the other depends on the specific application, level of accuracy desired, and time required to detect the microorganism [26]. To apply a rapid method to identify antimicrobial resistance, we developed an approach that consists of selecting bacterial resistance in a group of unknown species collected from an environmental source. The application of this method makes it easy to measure the abundance and knowledge of bacterial resistance, intrinsic and extrinsic, as well as information on the ecological state of health in relation to AMR. We aimed to investigate the green sea turtle as a reservoir of resistant bacteria, mainly because *C. mydas* is one of the most frequent sea turtle species in the Brazilian coastal regions and, consequently, under the intense impact of anthropic factors [13].

Cloaca and neck samples were collected from captured *C. mydas* at a beach in the metropolitan area of Rio de Janeiro. These samples were analyzed to study the transient microbiota and the gastrointestinal tract. Additionally, we investigated bacterial resistance to antimicrobials.

## 2. Results

Bacterial isolates from eleven turtles (11 neck and 11 cloaca samples) were grown from the 16 individuals with a mean curved carapace length (CCL) of 60.7 cm, ranging from 42.8 to 92 cm and an average weight of 39.6 kg, which measures the characteristics of juvenile green turtles. Five turtles’ cloaca and neck samples did not grow in the rich medium under the protocol conditions of the presence of antimicrobials. Nevertheless, samples from eleven turtles resulted in the growth of 130 bacterial isolates from neck and cloaca samples. These 130 bacterial isolates were further identified by MALDI-TOF or 16S rRNA (Appendix A).

### 2.1. Isolated Bacteria of Neck Samples

Seventeen bacterial isolates corresponded to Gram-negative Enterobacteriaceae, Morganellaceae, and 21 isolates of Gram-positive families of Enterococcaceae, Microbacteriaceae, and Promicromonosporaceae were identified (Table 1). Among the Gram-negative bacteria belonging to the Enterobacteriaceae family, *Citrobacter freundii* (*n* = 2), *Escherichia coli* (*n* = 1), *Salmonella* sp. (*n* = 1)*, Serratia marcescens* (*n* = 11), and *Serratia ureylitica* (*n* = 1), and the; Morganellaceae family, *Proteus mirabilis* (*n* = 1), were identified. Among Gram-positive, Enterococcaceae, *Enterococcus faecalis* (*n* = 8), Microbacteriaceae, *Microbacterium* spp. (*n* = 5), and Promicromonosporaceae, *Cellulosimicrobium cellulans* (*n* = 8) were found.

### 2.2. Isolated Bacteria of Cloaca Samples

Ninety-two bacterial isolates were identified, and among Gram-negative (*n* = 80), Enterobacteriaceae and Morganellaceae were found; for Gram-positive (*n* = 12), Enterococcaceae and Streptococcaceae were found (Table 2).

Among Enterobacteriaceae, the genera *Citrobacter* (*C. braaki*—*n* = 2, *C. freundii*—*n* = 20); *Klebsiella* (*K. oxytoca—n* = 1, *K. variicola—n* = 1); *Salmonella* sp. (*n* = 1), and *S. marcescens* (*n* = 1) were found. Among Morganellaceae, the genera *Morganella* and *Proteus* (*M. morganii—n* = 40 and *P. mirabilis* *n* = 14) were found. For the Gram-positive Enterococcaceae family, the genus *Enterococcus* (*E. faecalis—n* = 9, *E. hirae—n* = 2), and then for the Streptococcaceae family, the genus *Lactococcus* (*Lactococcus garvieae—n* = 1) were found.

### 2.3. Antimicrobial Bacterial-Resistant Isolates from Neck Samples

Three species resistant to gentamycin were found (*C. freundi*, *Microbacterium* spp., and *C. cellulans*); three species resistant to ciprofloxacin (*E. coli*, *P mirabilis*, and *C. cellulans*); and four species resistant to tetracycline (*Salmonella* sp., *S. marcescens*, *S. ureylitica,* and *E. faecalis*) were found (Table 3) No strain isolated from neck samples was classified as resistant to vancomycin.

### 2.4. Antimicrobial Bacterial Resistant Isolates from Cloaca Samples

Four species resistant to cephalothin were found (*C. braaki*, *K. oxytoca*, *K. variicola*, and *C. mirabilis*); two species resistant to tetracycline (*M. morganii* and *E. faecalis*); one species resistant to vancomycin (*E. faecalis*); and one species resistant to ciprofloxacin (*L. garvieae*) were found (Table 4). The eight tetracycline isolates of *E. faecalis* are resistant to azitromycin but susceptible to ampicillin, ciprofloxacin, rifampicin, and vancomycin.

## 3. Discussion

So far, no studies have evaluated the microbiota of intentionally captured, i.e., free-living, green turtles from Brazil. This study aimed to investigate the green sea turtle as a reservoir of resistant bacteria, mainly because *C. mydas* is one of the most frequent sea turtle species in Brazilian coastal regions and, consequently, under the intense impact of anthropic factors [13,27,28]. A group of 130 isolates of unknown species were collected from the cloaca and neck samples of green turtles. A greater cultured microbial diversity (gram-negative and gram-positive) was found in the cloaca (*n* = 92) compared with the neck (*n* = 38). Others [22,29,30] also used samples from sea turtles’ necks to represent the environment’s transient microbiota. 

Species resistant to antibiotics in the neck were found. Tetracycline had more resistant species, such as *E. faecalis*, *Salmonella* sp., *S. marcescens*, and *S. ureylitica*. A chronic intake of animal products with different tetracycline concentrations is known to be associated with dose-dependent changes in microbial populations obtained from human fecal samples [31]. For ciprofloxacin, three species were found to express resistance: *C. cellulans*, *E. coli*, and *P. mirabilis*. The genus *Cellulosimicrobium* is reported to exhibit resistance genes [32,33]. Ciprofloxacin-resistant and gentamicin-resistant *C. cellulans* were also isolated in neck samples. *C. cellulans* is commonly found in soil or surface waters and causes rare human infections [33]. Citrobacter is an opportunistic pathogen in some animals that causes severe aquaculture issues, leading to fish mortality. Findings of *Citrobacter* resistance in a neck sample suggest environmental contamination [34,35]. Gentamycin revealed resistance in *C. cellulans*, *C. freundii*, and *Microbacterium* spp. Interestingly, these bacteria are more related to the intestinal microbiota than the skin. In this case, the hypothesis could be raised that the environment where the turtles were was contaminated with sewage, since transient microbiota isolated from the neck are compatible with this hypothesis. Gentamicin is a class of antimicrobial commonly used in the production of plants to treat animals and humans [36]. This antibiotic has been found to have a high level of resistance in bacteria from the ESKAPE group (*Enterococcus faecium*, *Staphylococcus aureus*, *Klebsiella pneumoniae*, *Acinetobacter baumanii*, *Pseudomonas aeruginosa*, and *Enterobacter* spp.) [37].

*E. faecalis* was found to be resistant to tetracycline and azithromycin; this species accounts for most human enterococcal infections and may lead to lethal systemic diseases [38]. Resistant bacteria found in cloaca samples suggest that they are part of the gastrointestinal microbiota of sea turtles. For example, *P. mirabilis* was found to be resistant to cephalothin. Curiously, no report of cephalothin-resistant *P. mirabilis* was found [39]. Nevertheless, *P. mirabilis* naturally occurs in the gastrointestinal tract microbiota of sea turtles and other marine animals and is sometimes connected with illness [40]. This work showed antimicrobial resistance in *P. mirabilis*, indicating circulating resistant bacteria in sea turtle reservoirs, possibly due to the abusive use of antibiotics carried into the water by sewage. Detection of fecal contamination has evolved over the last 100 years but has mainly focused on coliforms, fecal coliforms, *E. coli*, or enterococci [41,42].

It was found that *M. morganii* was exclusively present in cloaca samples that had become resistant to tetracycline. This species has been identified as an opportunist pathogen, having acquired genes that confer resistance to tetracycline [31]. Furthermore, the cloaca samples showed the presence of *Lactococcus garvieae* which exhibited resistance to ciprofloxacin [43,44,45]. The presence of Ciprofloxacin-resistant *L. garvieae* in the gut of animals at the top of the food chain can have negative impacts on the environment and place sea turtles at risk.

This study observed no multi-resistant *E. faecalis* during the antimicrobial test. Our team plans to use molecular techniques to assess the microorganisms found in cloaca samples without bacterial cultivation or isolation. Through the metagenomic approach, the bacterial taxonomic groups and the resistant genes present in the intestinal tract of these sea turtles will be identified. Overall, this data will enhance our understanding of multiresistant bacteria in *C. mydas*.

The studies of the microbiota of sea turtles are improving the treatment and prevention of diseases in rehabilitation centers. Few studies are available to determine the microbiome of the gastrointestinal tract of stranded turtles and identify microorganisms that are potential human pathogens or that carry resistant genes [15,46,47]. Microbiological approaches to identifying resistant bacteria in sea turtle samples are widely used worldwide [16,22,23,48,49]. However, most studies investigate the resistance of the pathogen in other species of sea turtles, for example, the loggerhead, *Caretta caretta* [21,23,24,48,50]. Results from testing the wounds of injured *C. caretta* in the Mediterranean Sea revealed that 75% of the isolated bacteria displayed resistance to multiple drugs [51]. This suggests that resistant bacteria are present in sea turtles. Previous studies reported the presence of sea turtles in the Itaipú Beach area, which has an intense artisanal fishery and is influenced by Guanabara Bay water [52,53]. Furthermore, green sea turtles were observed feeding on discarded fish thrown into the seawater by the fishermen. The continuous contact with Guanabara Bay waters and this interaction with the rest of the fish can influence the microbiota and, consequently, the appearance of antimicrobial resistance [13,20].

Bacterial culture is used for resistance research in sea turtles, focusing on gram-negative bacteria of the Enterobacteriaceae family from rehabilitation centers [54], captured, or found stranded alive [50]. In addition, more bacteria from the Enterobacteriaceae family were observed, corroborating other studies [23,24]. The number of resistant isolates was more significant in the cloaca than the neck samples for most antimicrobials, reinforcing that the green turtle is a bioindicator for AMR, pollution, and anthropic impact, contributing to the One Health index’s performance [9,55].

## 4. Materials and Methods

### 4.1. Study Area

The Itaipú Beach is an oceanic beach located on the coastal plain of Niterói City, in Rio de Janeiro state, Brazil (22.97083° S, 43.04638° W). Appendix A shows the location on a map. This area is influenced by the waters flowing from Guanabara Bay, the second-largest estuary in the country, located in the metropolitan region of Rio de Janeiro. It presents intense port and industrial activity and houses one of the largest oil refineries in the country [56]. The beach is used by the population and tourists, mainly during vacations and in the summer, and has a strong presence of a local fishing community [57]. Itaipú Beach has previously been reported as a feeding, foraging, development, and residence area for juvenile green turtles by the Aruanã Project, which is dedicated to the conservation of sea turtles in Guanabara Bay and adjacent coastal regions [52,53].

### 4.2. Sample Collection

Sixteen cloaca and 16 neck samples (the latter representing the transient microbiota) were collected with sterile swabs streaked onto the nutrient agar slant and transported on ice to the laboratory for further analysis. The neck samples were collected using sterile swabs rubbed into the dorsal region of the neck. The samples were collected from 16 free-range wild green sea turtles between October and November 2019, in partnership with the Aruanã Project, under SISBIO Licenses 40873-6 and 71913-1. The sea turtles were captured intentionally through a beach trawl net. The individuals had a mean curved carapace length (CCL) of 60.7 cm, ranging from 42.8 to 92 cm, and a mean weight of 39.6 kg, which is characteristic of juvenile green turtles. The mean curved carapace width (CCW) was 61.5 cm, ranging from 40.4 to 82.1 cm (Appendix A). In particular, the CCL of the 16 captured turtles was >42 cm in all individuals.

### 4.3. Isolation of Bacteria

The samples on nutrient agar slant were cultured under aerobic conditions at 37 °C for 24–48 h. The cultures that showed growth were transferred to the tubes with broth. Furthermore, the samples were stored at −20 °C with the addition of 20% glycerol.Viability and antibiotic blind testing of bacterial cells stored frozen in glycerol were observed by seeding 50 μL on nutrient agar and MacConkey (BBL, Edwardsburgh/Cardinal, ON, Canada) medium with antibiotics—cephalothin (30 μg/mL), ciprofloxacin (5 μg/mL), gentamicin (10 μg/mL), tetracycline (30 μg/mL), and vancomycin (30 μg/mL) (Laborclin, RJ, Brazil) according to CLSI’s guidelines [39]. These antimicrobials were used for each sample. After the growth of colony-forming units (CFUs), five CFUs with different morphology and pigmentation were selected, and the CFU were placed in nutrient agar or MacConkey agar with antibiotics, respecting the colony’s medium, for isolation at 37 °C for 24 to 48 h (meta-phenotyping methodology described in detail in Figure 1). Once the colonies were isolated, identification could be made. Disk diffusion of antimicrobials into the solid culture medium was used for *E. faecalis* to test for multi-resistance using tetracycline, azithromycin, ampicillin, ciprofloxacin, rifampicin, and vancomycin [58].

### 4.4. MALDI-TOF MS Identification

Identification by Matrix-Assisted Laser Desorption/Ionization Time-of-Flight Mass Spectrometry (MALDI-TOF—matrix-assisted laser desorption ionization-time-of-flight) was carried out on the Microflex LT MS platform (Bruker Daltonics, Bremen, Germany). Samples were prepared as described elsewhere [59], and each isolate was analyzed in duplicate. The mass spectra obtained were compared with the references in the database using MALDI Biotyper 7.0 and Bruker Software for MALDI Imaging Data Analysis. The score values obtained were interpreted according to the manufacturer’s guidelines: ≥2.300 indicates reliable identification at the species level; 2.000–2.299 at the genus level (and likely identification for species); 1.700–1.999 solely at the genus level; and <1.699 is not considered reliable for identification. *Escherichia coli* DH5α was used as a quality control strain.

### 4.5. 16S rRNA Identification

DNA extraction was performed by the proteinase K method by adding phenol-chloroform (1:1). DNA quantification was performed using a spectrophotometer (Implen NanoPhotometer^®^, Westlake Village, CA, USA). PCR was performed using the primers V1V2- Forward 5′-AACACATGCAAGTCGAACG-3′ and V6V7V8-Reverse 5′-TCAACG TCTGAGGTTAGGCC-3′, which comprise a fragment of around 900 base pairs (bp) [60]. Amplification products were sequenced using Sequencing performed on the Applied Systems 3500 BigDye™ Terminator v3.1 Cycle Sequencing Kit (Applied Biosystems, Waltham, MA, USA) following the manufacturer’s protocol and run into the 3500 ABI Genetic analyzer (Thermofisher Scientific, Carlsbad, CA, USA) and sequences analyzed on Geneious v. 2022.1.1 to generate a consensus sequence. BLAST was used to compare the sequences with the nr database [61].

## 5. Conclusions

Green turtles in urban areas are constantly subject to contact with microorganisms coming from rivers and sewage, and turtles, through the food chain, accumulate these bacteria. This work is the first study of the phenotypic analysis of bacteria from free-living green turtles on Itaipú Beach, located in the metropolitan region of Rio de Janeiro. Bacteria found in neck samples are more related to the intestinal microbiota than the skin. One could raise the hypothesis that the environment where the turtles were was contaminated with sewage. *Proteus mirabilis* naturally occurs in the gastrointestinal tract microbiota of sea turtles and other marine animals and is sometimes connected with illness. Our work showed antimicrobial resistance in *P. mirabilis*, indicating circulating resistant bacteria in sea turtles as reservoirs, possibly due to the abusive use of antibiotics carried to the ocean by sewage. All antibiotics tested are of critical/high importance in human medicine, and the fact that we found many resistant bacteria may show the impact of humans on marine life, which may also represent a risk to the health of sea turtles. Identifying bacteria of medical importance that may possibly present resistance genes shows us that the environment is contaminated, thus creating a public health problem.

## Figures and Tables

**Figure 1 antibiotics-12-01268-f001:**
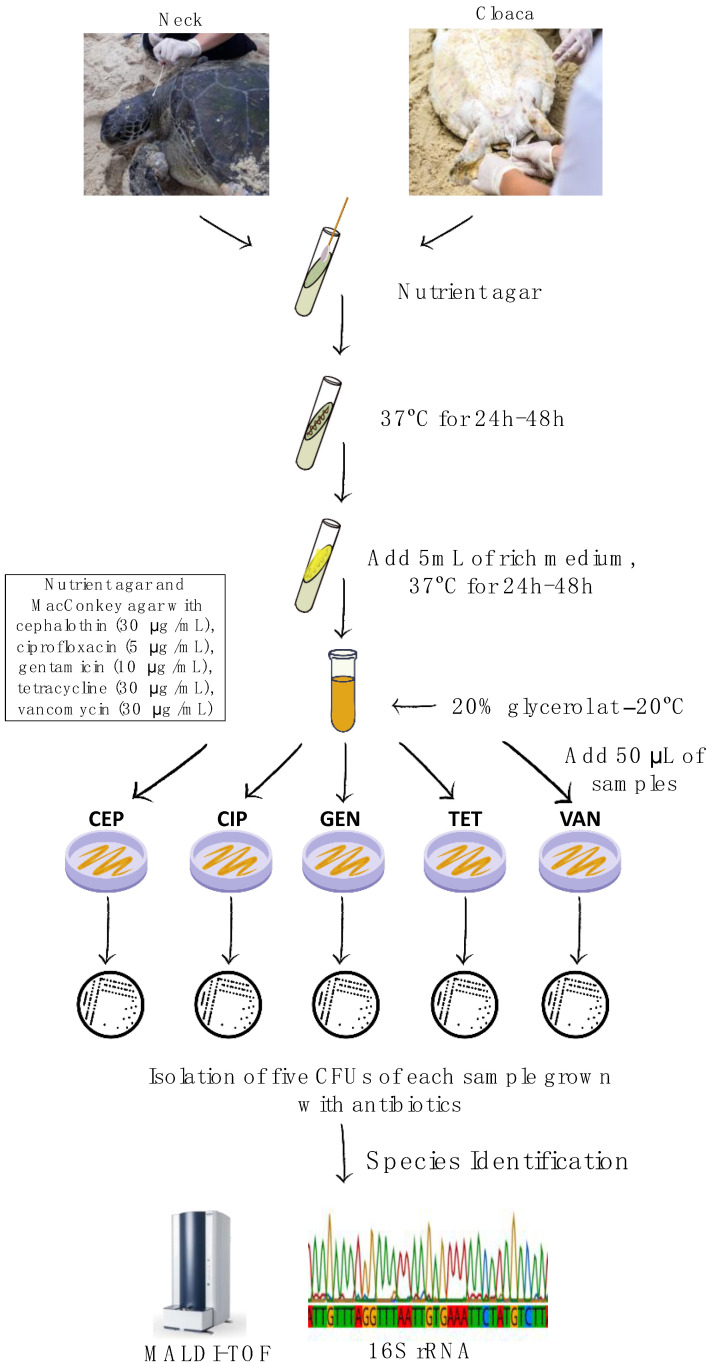
Diagram of the method. Abbreviation: CEP, cephalothin; CIP, ciprofloxacin; GEN, gentamicin; TET, tetracycline; VAN, vancomycin.

**Table 1 antibiotics-12-01268-t001:** Identification of isolated bacteria from neck samples.

Neck	Family	Genus	Species	Number of Isolates
Gram-negative	Enterobacteriaceae	*Citrobacter*	*C. freundii*	2
		*Escherichia*	*E. coli*	1
		*Salmonella*	*Salmonella* sp.	1
		*Serratia*	*S. marcescens*	11
			*S. ureylitica*	1
	Morganellaceae	*Proteus*	*P. mirabilis*	1
Gram-positive	Enterococcaceae	*Enterococcus*	*E. faecalis*	8
	Microbacteriaceae	*Microbacterium*	*Microbacterium* spp.	5
	Promicromonosporaceae	*Cellulosimicrobium*	*C. cellulans*	8

**Table 2 antibiotics-12-01268-t002:** Identification of isolated bacteria from cloaca samples.

Cloaca	Family	Genus	Species	Number of Isolates
Gram-negative	Enterobacteriaceae	*Citrobacter*	*C. braaki*	2
			*C. freundii*	20
		*Klebsiella*	*K. oxytoca*	1
			*K. variicola*	1
		*Salmonella*	*Salmonella* sp.	1
		*Serratia*	*S. marcescens*	1
	Morganellaceae	*Morganella*	*M. morganii*	40
		*Proteus*	*P. mirabilis*	14
Gram-positive	Enterococcaceae	*Enterococcus*	*E. faecalis*	9
			*E. hirae*	2
	Streptococcaceae	*Lactococcus*	*L. garvieae*	1

**Table 3 antibiotics-12-01268-t003:** Antimicrobial-resistant isolates from neck.

Neck	Family	Genus	Species	Number of Resistant Isolates (Antimicrobial)
	Enterobacteriaceae	*Citrobacter*	*C. freundii*	1 (GEN)
Gram-negative		*Escherichia*	*E. coli*	1 (CIP)
		*Salmonella*	*Salmonella* sp.	1 (TET)
		*Serratia*	*S. marcescens*	3 (TET)
			*S. ureylitica*	1 (TET)
	Morganellaceae	*Proteus*	*P. mirabilis*	1 (CIP)
Gram-positive	Enterococcaceae	*Enterococcus*	*E. faecalis*	6 (TET) *
	Microbacteriaceae	*Microbacterium*	*Microbacterium* spp.	5 (GEN)
	Promicromonosporaceae	*Cellulosimicrobium*	*C. cellulans*	2 (CIP) 6 (GEN)

* Susceptibility test showed that the six isolates are resistant to tetracycline and azitromycin but susceptible to ampicillin, ciprofloxacin, rifampicin, and vancomycin.

**Table 4 antibiotics-12-01268-t004:** Antimicrobial bacterial resistant isolates from the cloaca.

Cloaca	Family	Genus	Species	Number of Resistant Isolates (Antimicrobial)
	Enterobacteriaceae	*Citrobacter*	*C. braaki*	2 (CEP)
Gram-negative		*Klebsiella*	*K. oxytoca*	1 (CEP)
			*K. variicola*	1 (CEP)
	Morganellaceae	*Morganella*	*M. morganii*	2 (TET)
		*Proteus*	*P. mirabilis*	1 (CEP)
Gram-positive	Enterococcaceae	*Enterococcus*	*E. faecalis*	8 (TET) * 1 (VAN)
	Streptococcaceae	*Lactococcus*	*L. garvieae*	1 (CIP)

* Susceptibility test showed that the eight isolates from cloaca are resistant to tetracycline and azitromycin but susceptible to ampicillin, ciprofloxacin, rifampicin, and vancomycin.

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
