# Peer review of "Antimicrobial-Resistant Bacteria from Free-Living Green Turtles (Chelonia mydas)"

_antibiotics, 2023, doi:10.3390/antibiotics12081268_

Round 1

Reviewer 1 Report

the paper is well articulated and takes up other studies conducted on other species of sea turtles (mostly caretta caretta) in which antibiotic resistance has been reported, especially in marine areas of the Mediterranean. I believe that this work can help improve the scientific data available on these animals, especially for their value as marine bioindicators of antibiotic resistance. I recommend increasing the two articles suggested in the text to complete what is reported by the authors. I recommend reviewing English, especially trying to replace some verbs in the past tense. 

minor revision, can be improved. 

Author Response

Dear Reviewer

We carefully revised your recommendations and made point-by-point changes to the article. All issues have been resolved, making the manuscript better for readers of Antibiotics.

Comments and Suggestions for Authors 

The paper is well articulated and takes up other studies conducted on other species of sea turtles (mostly caretta caretta) in which antibiotic resistance has been reported, especially in marine areas of the Mediterranean. I believe that this work can help improve the scientific data available on these animals, especially for their value as marine bioindicators of antibiotic resistance. I recommend increasing the two articles suggested in the text to complete what is reported by the authors. I recommend reviewing English, especially trying to replace some verbs in the past tense.  

Response: We added the two references as suggested for a broader subject explanation.   

I recommend reviewing English, especially trying to replace some verbs in the past tense.  

Response: We reviewed the text and replaced it with past tense where it applies. 

Attached is the revised manuscript with track changes and a clean version in Word and pdf files. Information requested by reviewers is highlighted in yellow. Best regards

Reviewer 2 Report

Dear authors,

you presented the results of the investigation of the finding of resistant bacteria, isolated from the free-living sea turtles, captured in Brazil.

The topic of this research is interesting and the results could be useful for many other researchers, but unfortunately the text itself is written quite incomprehensible and makes manuscript difficult to understand.

In particular, attention should be paid to the chapters Introduction, Results and Discussion. There are multiple sentences in the manuscript that need revision, but it is difficult to mention it as there is no corresponding line numbers.

Specific comments:

- please uniform wording anthropic or anthropogenic in the text 

- neck samples - please clearly state whether you mean skin swabs from the neck area?

Last paragraph in the Introduction - please provide reference for the approach you developed for the detection of AMR in unknown bacterial species.

Under chapter 2.4. you mention resistance to azithromycin, but later on in the Material and methods you didn't mentioned this antibiotic (also, the asterisk under the table 4 mention different antimicrobials, that are not later investigated).

Sentences in Discussion, as "Tetracyclin had more resistant species..." should be changed and revised.

Chapter 4.3. - by "growth of CFU" you actually mean growth of colony forming bacteria?

Figure 1 - do you miss one step, related to addition of 50 microliters of samples to the nutrient or MacConkey agar?

Sentences are not clear enough, very difficult to understand, major revision needed.

Author Response

Dear Reviewer

We carefully revised your recommendations and made point-by-point changes to the article. All issues have been resolved, making the manuscript better for readers of Antibiotics.

Reviewer 

Comments and Suggestions for Authors 

Dear authors, 

you presented the results of the investigation of the finding of resistant bacteria, isolated from the free-living sea turtles, captured in Brazil. The topic of this research is interesting, and the results could be useful for many other researchers, but unfortunately the text itself is written quite incomprehensible and makes the manuscript difficult to understand.  difficult to understand.In particular, attention should be paid to the chapters Introduction, Results and Discussion. There are multiple sentences in the manuscript that need revision, but it is difficult to mention it as there are no corresponding line numbers. no corresponding line numbers. 

 Specific comments: 

- please uniform wording anthropic or anthropogenic in the text in the text  

Response: The term anthropic has been standardized. 

- neck samples - please clearly state whether you mean skin swabs from the neck area? 

Response: We added this information in Materials and Methods /4.2 Sample collection   

Last paragraph in the Introduction - please provide reference for the approach you developed for the detection of AMR in unknown bacterial species. 

Response: We added a full detailing of the method used, please refer to that section for better comprehension of the technique. 

Under section 2.4. you mention resistance to azithromycin, but later on in the Material and methods you didn't mentionthis antibiotic (also, the asterisk under table 4 mention different antimicrobials, that are not later investigated). different antimicrobials, that are not later investigated). 

Response: We used different antimicrobials to test multi-resistance for E. faecalis only. We added the method used for E. faecalis. 

Sentences in Discussion, as "Tetracyclin had more resistant species..." should be changed and revised. 

Response: We fixed the sentences.  

Chapter 4.3. - by "growth of CFU" you actually mean growth of ‘colony forming bacteria’? growth of colony forming bacteria? 

Response: We use CFU for “colony-forming units”.  

Figure 1 - do you miss one step, related to addition of 50 microliters of samples to the nutrient or MacConkey agar? 

Response: We added this step to Figure 1. 

Comments on the Quality of English Language 

Sentences are not clear enough, very difficult to understand, major revision needed. 

Response: we extensively revised the English grammar of the manuscript. 

  Attached is the revised manuscript with track changes and a clean version in Word and pdf files. Information requested by reviewers is highlighted in yellow. Best regards

Reviewer 3 Report

Dear Authors,

Greetings!

The manuscript “Antimicrobial-resistant bacteria from free-living green turtles (Chelonia mydas)” presents data related to antimicrobial-resistant bacteria present in the neck and cloaca of this marine pollution’s bioindicator.

However, prior to publishing the study in “Antibiotics” some adjustments are necessary to enhance quality and readers’ comprehension. 

The dot at the end of the title needs to be removed.

Regarding the species name (Chelonia mydas) throughout the manuscript, after mentioning it for the first time, it is recommended to use only the abbreviated form (C. mydas) in the following mentions.

Regarding “Abstract”, it would be interesting to remove the repetition of “we”, using passive voice instead, for example. The italics can be removed from “antibiotic resistance”. Regarding the terms used to refer to the cloacal region, cloaca seems to be the correct one. The word “cloacal” is an adjective and not a noun.

When it comes to “Introduction”, some aspects need to receive attention such as: 1) explaining the importance of analyzing neck and cloaca and what can be informed by the results from each of these areas; 2) informing the dimension of the problem associated with the study: resistance to antimicrobial drugs causes how many deaths worldwide? There are economic aspects associated with trying to treat individuals infected by resistant bacteria?; 3) addressing the action mechanism of antibiotics studied in the manuscript and their chemical classification; 4) mentioning main methods of identifying resistant organisms, highlighting the ones applied (MALDI-TOF and 16S rRNA); 5) informing a reference to the sentence “Chelonia mydas is one of the most frequent sea turtle species in Brazilian coastal regions and, consequently, under the intense impact of anthropic factors”. 

Regarding “Results”, it is interesting to substitute “We” by passive voice, to make the text impersonal and more formal. “22 neck and cloaca samples” would be more clear if presented as “11 neck and 11 cloaca samples”. The number 92 does not possess the word “and” in it; it is necessary to fix that. The study lacks a table presenting the results regarding which species were identified using MALDI-TOF and which ones required 16S rRNA analysis. Regarding the last ones, it is also necessary to inform readers how many pb each amplified fragment amplified presented, and homology data after alignment.   

When it comes to “Discussion”, the sentence “For ciprofloxacin, we found three species expressing”… is missing a “:” between “resistance” and “C. cellulans”. The sentence “Curiously, no report of cephalothin-resistant P. mirabilis [27]” seems to lack the end; it is necessary to dedicate attention to that. The word “studies” seems to be missing in this sentence: “Previous reported the presence…”. It is necessary to elaborate a more detailed discussion addressing diseases associated with each bacterium studied as same as if their resistant version was already reported, referencing it. It is also necessary to reunite the information and present to readers which species are known as being present in turtles’ microbiota, in order to verify if the species from resistant organisms are naturally present in the reptile’s body or not.

Regarding “Materials and Methods”, it is missing a subsection dedicated to presenting details related to equipment and reagents/culture media/antibiotics used. It is important to allow readers to repeat the procedures if they want to.  A map presenting the study area would be an interesting addition. In subsection 4.3 it is possible to notice that the study lacks regular bacteria to consist in drug susceptibility control.

When it comes to the “Conclusion”, it lacks mentions of the relevance of these turtles as bioindicators. It also does not present any future perspective.

Minor editing of English language required; a revision performed by a native or a professional service is necessary to adjust aspects such as:

1)      Maintaining a standard when it comes to style, such as in the sentence “Usually, the methodology includes collecting samples from sea turtles, isolating, and identifying bacteria, and posterior use of susceptibility tests [13–16]”.

2)      Choosing more suitable words in some sentences such as “The genus Cellulosimicrobium is shown to be carrying” and “… isolated, identification could be made”.

3)      Adjusting words such as the term to refer to the cloacal area; cloaca seems to be the correct one. The word “cloacal” is an adjective and not a noun. It is necessary to fix this confusion.

Author Response

Dear Reviewer

We carefully revised your recommendations and made point-by-point changes to the article. All issues have been resolved, making the manuscript better for readers of Antibiotics.

Following, we outline the adjustments made, point by point.

Comments and Suggestions for Authors 

Dear Authors, 

Greetings! 

The manuscript “Antimicrobial-resistant bacteria from free-living green turtles (Chelonia mydas)” presents data related to antimicrobial-resistant bacteria present in the neck and cloaca of this marine pollution’s bioindicator. However, prior to publishing the study in “Antibiotics” some adjustments are necessary to enhance quality and readers’ comprehension.  

  

The dot at the end of the title needs to be removed. 

Response: the dot was removed. 

  

Regarding the species name (Chelonia mydas) throughout the manuscript, after mentioning it for the first time, it is recommended to use only the abbreviated form (C. mydas) in the following mentions.  

Response: we made the necessary changes accordingly. 

Regarding “Abstract”, it would be interesting to remove the repetition of “we”, using passive voice instead, for example. The italics can be removed from “antibiotic resistance”. Regarding the terms used to refer to the cloacal region, cloaca seems to be the correct one. The word “cloacal” is an adjective and not a noun. 

Response: we made the necessary changes accordingly. 

When it comes to “Introduction”, some aspects need to receive attention such as:  

1) explaining the importance of analyzing neck and cloaca and what can be informed by the results from each of these areas;  

Response: To explain the importance of the samples, we introduced the sentence: “Samples were collected from the neck and cloaca of captured C. mydas to characterize the transient microbiota and the gastrointestinal tract, respectively”. 

2) informing the dimension of the problem associated with the study: resistance to antimicrobial drugs causes how many deaths worldwide? Are there economic aspects associated with trying to treat individuals infected by resistant bacteria? 

Response: We added information about the problem associated with our study: highlighted in yellow. 

3) addressing the action mechanism of antibiotics studied in the manuscript and their chemical classification;  

Response: We added the text according to the suggestion: highlighted in yellow. 

4) mentioning main methods of identifying resistant organisms, highlighting the ones applied (MALDI-TOF and 16S rRNA);  

Response: We added the text: highlighted in yellow. 

5) informing a reference to the sentence “Chelonia mydas is one of the most frequent sea turtle species in Brazilian coastal regions and, consequently, under the intense impact of anthropic factors”.  

Response: We added the reference accordingly. 

  

Regarding “Results”, it is interesting to substitute “We” by passive voice, to make the text impersonal and more formal. “22 neck and cloaca samples” would be more clear if presented as “11 neck and 11 cloaca samples”. The number 92 does not possess the word “and” in it; it is necessary to fix that.  the word “and” in it; it is necessary to fix that.  

Response: We made the necessary changes accordingly. 

The study lacks a table presenting the results regarding which species were identified using MALDI-TOF and which ones required 16S rRNA analysis. Regarding the last ones, it is also necessary to inform readers how many pb each amplified fragment presented, and homology data after alignment.    

Response: We included a supplemental table (Table S1) indicating the ones identified by Maldi-TOF, 16S rRNA sequencing, or both, including the size of the query sequence and identity. 

When it comes to “Discussion”, the sentence “For ciprofloxacin, we found three species expressing”… is missing a “:” between “resistance” and “C. cellulans”. ellulans”.  

Response: The “:” was added. 

The sentence “Curiously, no report of cephalothin-resistant P. mirabilis [27]” seems to lack the end; it is necessary to dedicate attention to that.  

Response: We completed the sentence. 

The word “studies” seems to be missing in this sentence: “Previous reported the presence…”.   

Response: We modified the phase. 

It is necessary to elaborate a more detailed discussion addressing diseases associated with each bacterium studied as same as if their resistant version was already reported, referencing it. It is also necessary to reunite the information and present to readers which species are known to be present in turtles’ microbiota, in order to verify if the species from resistant organisms are naturally present in the reptile’s body or not.  order to verify if the species from resistant organisms are naturally present in the reptile’s body or not.  

Response: The discussion session was enriched with information about turtle gut microorganisms and antimicrobial resistance 

  

Regarding “Materials and Methods”, it is missing a subsection dedicated to presenting details related to equipment and reagents/culture media/antibiotics used. It is important to allow readers to repeat the procedures if they want to.   

Response: We added the information accordingly. 

A map presenting the study area would be an interesting addition.  

Response: We added a map presenting the study area as a supplemental Figure S1. 

In subsection 4.3 it is possible to notice that the study lacks regular bacteria to consist in drug susceptibility control.   

Response: We added the test performed for E. faecalis in methods. 

  

When it comes to the “Conclusion”, it lacks mentions of the relevance of these turtles as bioindicators. It also does not present any future perspective. 

Response: Discussion on multi-resistance and metagenomic analyses. 

Comments on the Quality of English Language 

Minor editing of English language required; a revision performed by a native or a professional service is necessary to adjust aspects such as: 

1)      Maintaining a standard when it comes to style, such as in the sentence “Usually, the methodology includes collecting samples from sea turtles, isolating, and identifying bacteria, and posterior use of susceptibility tests [13–16]”.  

2)      Choosing more suitable words in some sentences such as “The genus Cellulosimicrobium is shown to be carrying” and “… isolated, identification could be made”.  

3)      Adjusting words such as the term to refer to the cloacal area; cloaca seems to be the correct one. The word “cloacal” is an adjective and not a noun. It is necessary to fix this confusion. 

Response: We have improved the English grammar and style. 

Attached is the revised manuscript with track changes and a clean version in  pdf. Information requested by reviewers is highlighted in yellow.

Best regards

Round 2

Reviewer 2 Report

Dear authors,

thank you for taking into consideration all the comments given by the reviewer. 

Best regards

Still some minor editing needed

Reviewer 3 Report

Manuscript improved as suggested.